# Quality Assurance and Cost Reduction in Histopathology Laboratories Using Tissue Microarrays

**DOI:** 10.3390/vetsci10040280

**Published:** 2023-04-06

**Authors:** Daniela Gologan, Alina Elena Ștefan, Manuella Militaru, Andreea Cristina Sanda, Suzana Arjan, Sorin Mușat, Matthew Okerlund Leavitt, Raluca Stan

**Affiliations:** 1Department of Organic Chemistry, Doctoral School, Faculty of Chemical Engineering and Biotechnologies, Politehnica University, 011061 Bucharest, Romania; daniela.gologan@yahoo.com (D.G.);; 2Department of Research and Development, Themis Pathology SRL, 077190 Bucharest, Romania; alina_stefan5@yahoo.com (A.E.Ș.); andreeacristina.sanda@yahoo.com (A.C.S.); suzy_jesus@yahoo.com (S.A.); 3Department of Pathology, Doctoral School, Faculty of Veterinary Medicine, University of Agronomical Sciences and Veterinary Medicine, 011464 Bucharest, Romania; militmanuella@yahoo.com; 4Department of Research and Development, LUMEA Inc., Lehi, UT 84043, USA; matt@lumea.net

**Keywords:** tissue microarray, quality assurance, cost-reduction, tissue processing protocol, multiplexing

## Abstract

**Simple Summary:**

The current need to reduce costs in research and diagnostic histopathology is fueling the development of new technologies. Tissue microarray (TMA) technology facilitates the simultaneous preparation of multiple tissue samples, their expeditious analysis, and a much more accurate diagnostic. The sectionable support matrix with multiple receptacles (for placement of core tissue samples collected from “donor” paraffin blocks) developed in our laboratory reduced the total histopathological costs (time and consumables) by 71%, while archiving and storage costs were reduced by 96%. We also observed that the quality of the “donor” paraffin blocks has an important influence on successfully multiplexing tissue samples. Despite the constant pressure for cost cutting and fast turnaround times, appropriate tissue processing should never be overlooked.

**Abstract:**

In the context of cost increases of both labor and consumables, cheaper and faster histopathology methods are needed. We implemented in our research laboratory the use of tissue microarrays (TMA) for the parallel processing and analysis of tissue samples. In this study, we used seven pre-processed, paraffinated biomimetic sectionable support matrices serving as “recipient” paraffin blocks to embed a total of 196 tissue cores from formalin-fixed paraffin-embedded tissue samples (serving as “donor” paraffin blocks) from seven different rabbit organs. These tissue samples were obtained using four different processing protocols: two 6 h protocols with xylene as the transition solvent, and two using butanol instead (one 10 h in duration and the other 72 h long). While the samples from protocols 1 and 2 (with xylene) quite regularly generated peeling of some of the cores from the slides (most likely because of substandard paraffin infiltration), butanol processing performed flawlessly for both processing protocols. Our proposed technique of using TMAs in the research laboratory brings with it a significant reduction in time and consumable costs (up to 77 and 64%, respectively), but also new challenges for all the upstream processes.

## 1. Introduction

The cost of medical histopathology includes and reflects all the consumables used upstream of the actual analysis/diagnostic, the salaries of the employees who perform these services, and the resources used (electric power, water, etc.). In most laboratories around the world, the histological protocols used for tissue processing are laborious, tedious, expensive, and with significant malpractice risks (small tissue fragments can be damaged or even lost). More and more studies are focusing on improving these procedures for better diagnosis and at the same time trying to find alternative protocols for reducing the costs of labor and materials. Numerous studies suggest that multiplexing methods can reduce costs by almost half and can also improve the efficiency of the diagnosis [1,2,3]. Tissue microarray (TMA) technology facilitates the parallel analysis of multiple tissue samples, giving us a tool for increasing the efficiency, standardization, and accuracy of many histological techniques in both clinical and research laboratories [1,4]. Two multiplexing methods are already used in clinical diagnosis: a method was developed in 2013 by S. Mușat, represented by the BxChip™ (LUMEA Inc., Lehi, UT, USA), the first clinical TMA [5], and a second method was developed by W.P. Williamson in 2014, represented by the Tissue-Tek^®^ Paraform^®^ Tissue Orientation Gels (Sakura, Alphen aan den Rijn, The Netherlands) [6].

BxChip™ is a biomimetic organic polymer, sectionable after histological processing and paraffin infiltration, with similar properties to human tissue [7]. BxChip™ incorporates linear biopsies, and it is available in a range of different gauge sizes. The design of this matrix allows the surgeon, interventional radiologist, or nurse in the operating room to horizontally arrange the freshly collected tissue cores with the help of the biopsy needle in grooves of appropriately sized geometries for all major types of tru-cut diagnostic procedures. The chemical composition of the BxChip™ allows its histological processing (dehydration, clarification and paraffin infiltration) simultaneously with the biopsies within it, so that they fuse with the matrix. Alternatively, pre-fixed tissue biopsies can be multiplexed within the BxChip™ in the Pathology Laboratory. BxChip™ have been available for clinical diagnostic since 2012 and was used mainly for prostate and breast biopsies [8,9], eliminating completely the fragmentation, deformation, or loss of orientation, etc., of the diagnostic tissue sample.

Tissue-Tek^®^ Paraform^®^ Tissue Orientation Gels are ready-to-use semi-solid hydrogels intended to be inserted into Tissue-Tek^®^ Paraform^®^ Cassettes and are claimed to preserve the orientation of very tiny tissues during the grossing, processing, sectioning and also facilitating imaging of the resulting stained slides. Tissue Orientation Gels are available in five configurations: biopsy gel, 2 lanes, and 1, 2, and 3 mm punch gels [10]. The disadvantage of this method is the lack of an identification code for the patient or the number of excised biopsies [4]. 

Several studies have demonstrated the superiority of these sectionable matrices compared with classical histological methods for analysis of core biopsies.

In 2014, Radavoi et al. compared prostate biopsies collected directly into the BxChip™ (Biopsy Chip) with the biopsies examined before the introduction of the multiplexing method. The multiplexing method not only significantly decreased the costs, but also increased the yield of diagnostic tissue on the microscope slides [11].

In 2016, Wojno et al. observed that using BxChip™ for the tissue processing of prostatic biopsies led to an increase in the cancer detection rate from 49.5% (standard method) to 58.8% [12].

Murugan et al. (2019) observed reduced tissue fragmentation and increased efficiency of biopsy diagnosis by using the BxChip™ multiplexing method [1]. The BxChip™ processing method allowed a reduction of up to 4 times in the time required to prepare the samples. Moreover, the space storage of BxChip™ blocks (*n* = 48) vs. standard tissue blocks (*n* = 288) was reduced 6-fold [1].

In 2022, Tomosoiu et al. reported that the precision of diagnosis in prostate cancer (concordance between needle biopsy and radical prostatectomy Gleason scores of adenocarcinoma) was superior when using the BxChip™ for collecting and processing core biopsies vs. conventional methods (69% vs. 43%, respectively) [13].

TMAs are already accepted as valuable tools in biomedical research and experimental pathology due to their ability to consolidate formalin-fixed paraffin-embedded tissue samples in just a few paraffin blocks for cost reduction as well as for reducing experimental variability. The tissue cores from the “donor” paraffin block can be extracted using skin punches of different inner diameters [14] or by using inexpensive, improvised instruments (mechanical pencil tips) [15]. 

In his comprehensive review from 2014, Vogel reviewed the studies published between 1965 and 2013 that described various TMA techniques aimed at cost reduction and/or the generation of positive controls [16]. 

In a research laboratory, a TMA can be constructed in three ways: oDuring sectioning of the “donor” paraffin block, paraffin sections are arrayed directly on slides (method 1) [17];oEmploying “receptor” blocks: paraffin blocks with pre-drilled holes serving as receptacles for the “donor” tissue cores (method 2) [16];oA third method consists of using a paraffinated support matrix—agar plates with embedded fixed tissue rods [18]—or plant-based matrices [19]).Paraffinated support matrices can have predetermined designs and are used as receivers for the donor biopsies. The loaded support matrices are further embedded in a paraffin block.Method 1 and method 2 are time consuming, and there is a high risk of losing the cores during embedding [14,20,21,22]. 

In 2023, Stefan et al. developed a pre-paraffinated biomimetic material (dehydrated, clarified and paraffin-infiltrated biomimetic matrix) used to incorporate fresh tissue samples. The assembly fresh samples/paraffinated matrix is run through another round of dehydration, clarification, and infiltration with paraffin to generate the final TMA. These matrices demonstrated a remarkably stable structure, allowing trouble-free serial microtome sectioning [23].

Regardless of the method employed, once constructed, the paraffinated TMA composite block can be used to obtain hundreds of serial sections to be used further for hematoxylin-eosin staining or ancillary analyses (immunohistochemistry, fluorescence in situ hybridization, etc.). 

Many research applications require the testing/quantification of multiple markers in the same tissue sample or the testing of a single marker in a very large number of tissue samples [24]. For example, Jensen et al., in 2014, validated 175 antibodies using the TMA method. By employing this multiplexing technique instead of individual slides, he decreased the costs from USD 196,437.50 to USD 7857.50 [25].

These complex analyses require adequate stabilization/preservation of the tissue samples. Proper collection, fixation, dehydration, and infiltration of the samples are required to stabilize and preserve the tissue for long periods (years, decades). Immunohistochemical and nucleic acid-based assay results can be vitiated when dealing with paraffin blocks stored for a very long time. It is estimated that with every decade of storage, the quantity of nucleic acids is reduced with 5 to 50% [26]. For example, it was observed that estrogen and progesterone receptor expression levels were reduced in 10-year-old samples, while human epidermal growth factor receptor 2 and chromosome enumeration probe 17 signal intensities decreased proportionally with the age of the paraffin blocks [26]. Unfortunately, due to the lack of standardization of pre-analytical methods, tissue processing protocols can vary from 1–2 h to 24 (or more) hours, depending on the type and size of the sample. Unsatisfactory diffusion of solvents into the tissue sample can result in residual water in paraffin blocks of up to 1.5–3% [27]. The importance of choosing appropriate processing protocols cannot be overemphasized. Too much or insufficient time in dehydrant solutions will dramatically influence the quality of the final paraffin blocks. Tissues shrink significantly if exposed for too much time to a high concentration of alcohol or if the temperature of the solutions is too high. The paraffin blocks will section with difficulty, and the tissue sections will present artifacts such as holes, scratches, or even breaks. Residual water in the processed tissue samples can generate cracks or depressions in the paraffin block when exposed to air. Variations in the environmental conditions, such as the presence of water (endogenous and exogenous), exposure of the cut surfaces to air (oxidation), storage temperature, etc., all can influence the quality of archived paraffin blocks [28,29].

Tissue microarrays can facilitate the standardization of different staining techniques or molecular assays so that results are more reproducible, and the costs associated with performing downstream applications on many individual samples arrayed in a single paraffin block are reduced [14,30,31].

This study aims to implement a new method of multiplexing different types of paraffinated tissue samples by using a paraffinated support matrix. The support matrix was obtained from a biomimetic sectionable material developed by Stefan et al. in 2023 [23] and designed by us with a new geometry. We also analyzed the cost effectiveness of the method as well as the influence of the upstream processing protocols used for the “donor” blocks on the quality of the resulting tissue microarray.

## 2. Materials and Methods

### 2.1. Tissue Samples

For this study, we used New Zealand rabbits provided by the “Cantacuzino” National Military-Medical Institute for Research and Development. The rabbits were euthanized with an overdose of anesthetic (T61) administered intravenously. The animal study protocol was approved (approval code CE/231 from 19 June 2020) by the Ethics Commission of “Cantacuzino” National Military-Medical Institute for Research and Development, Bucharest, Romania. The tissue samples were harvested by a veterinary surgeon at the “Băneasa” Animal Facility in the Preclinical Testing Unit (part of Cantacuzino National Military-Medical Institute for Research and Development and authorized). All tissue samples were fixed immediately in 10% NBF (neutral buffered formalin).

The following organs were collected: liver, kidney, spleen, jejunum, skeletal muscle, testes, and ear. For each specimen, 4 pairs of biopsies of the same size were obtained, fixed in 10% neutral buffered formalin for 24 h, and processed with fresh reagents with 4 tissue processing protocols, resulting in a total of 196 paraffin blocks. Protocol 1 and Protocol 2 are used routinely in histology laboratories, while Protocols 3 and 4 were developed in our laboratory. Dehydration in all protocols was conducted with increasing concentrations of ethanol (70%, 80%, 95%, 100%); the main difference between these 2 groups of protocols is the clearing solvent used: xylene (for Protocols 1 and 2), which is the solvent most widely used in histology because it displaces alcohol rapidly, and *n*-butanol (for Protocols 3 and 4). Since it is partially miscible with water, butanol is a very good intermediate between alcohol and paraffin. For each processing protocol, the conditions of temperature and time varied. Protocol 1: 6 h at 37 °C; Protocol 2: 6 h at room temperature; Protocol 3: 10 h at 37°C; Protocol 4: 3 days at room temperature. For Protocols 3 and 4, we allotted more time for clarification and infiltration since butanol is a slower-acting dehydrant. For Protocol 4, two changes of *n*-butanol were performed at 24 h. 

The biopsies were embedded in paraffin (Leica Biosystems, Wetzlar, Germany) at 60 °C, under vacuum, using regular histological cassettes (“donor” paraffin blocks) and sectioned at 5 µm on a Leica 2235 rotary microtome (Leica Microsystems Ltd., Shanghai, China). Standard microscope slides (26 × 76 mm) (Bio-Optica Milano S.p.a, Milano, Italy) were stained with hematoxylin-eosin and cover-slipped (24 × 50 mm) (Bio-Optica Milano S.p.a, Milano, Italy) for microscopic examination. After sectioning, the resulting paraffin blocks were used as “donor” paraffin blocks for tissue microarrays (TMA).

### 2.2. Tissue Microarrays

To obtain paraffinated support matrices, 5 mm thick slices of a biomimetic material were used (Themis Pathology S.R.L, Bucharest, Romania), fixed for 24 h with 10% neutral buffered formalin, dehydrated with progressive concentrations of ethanol, cleared with a transitional solvent, and infiltrated with paraffin. 

The biomimetic material contained hydrophilic components such as proteins, polymerizable carbohydrates, surfactants, pigments, etc., and hydrophobic components such as saturated fatty acids. After processing protocol, the material reached a thickness of 2 mm. The resulting support matrices (50 × 37 × 2 mm) were etched by laser engraving to create 28 receptacles. The receptacles are holes with a 6 mm diameter, arranged in 4 rows and 7 columns. We decided to customize each TMA with 28 receptacles to include all the samples derived from one single type of organ on a paraffin block, resulting in 7 “receptor” blocks. Each “receptor” block contained a tissue microarray with a predefined design for easy identification of the individual tissue samples. On each TMA, we also engraved information about the project number (#7–20), tissue type (liver, kidney, spleen, jejunum, skeletal muscle, testes, and ear), identification number of the experimental animal (columns 1–7), and processing protocols used (Figure 1). 

Prior to starting the multiplexing of the tissue samples, the “donor” blocks were warmed at 35 °C in an incubator for 30 min. This prior warming of the blocks keeps the paraffin soft, and the force required for punching out the donor cores is reduced, eliminating the risk of cracking the paraffin blocks. A 6 mm disposable biopsy punch (Kai Industries co., ltd., Seki, Japan) was used to extract the tissue cores from the “donor” blocks. The selected cores were inserted manually into the receptacles of the recipient support matrix, according to the corresponding location indicated by the coordinate numbers (Figure 2). The tissue final microarray was placed in the mold face down so that when sectioning the paraffin block, the entire information engraved on the TMA could be read.

Molten paraffin was poured slowly, avoiding the formation of air bubbles into the metal mold, and maintained on the hot plate of the embedding station for an additional 5 min. The mold was then cooled, and once the paraffin had solidified, the paraffin block (Figure 3A) was removed from the mold. Sectioning was performed at 5 µm on a Leica 2235 rotary microtome, the paraffin sections were floated on a water bath at 42 °C, flattened, and affixed on positively charged glass slides (CellPath Ltd., Newtown, UK), and stained with hematoxylin-eosin (Figure 3B).

### 2.3. Cost Analysis

A comparative cost analysis of the classic method vs. our TMA multiplexing method was conducted by testing 4 tissue processing protocols using 49 samples for each protocol. Seven 28-core TMAs were created (for seven types of tissue: liver, kidney, spleen, jejunum, skeletal muscle, testes, and ear). The study included only the costs incurred after preparing the donor blocks—the cost of the customized support matrix, all the consumables used to obtain the staining slides from the “donor” and the “receptor” block, and histotechnologist time. The cost of the support matrix was the total cost (i.e., it includes the biomimetic material, solutions used to process the material, and laser engraving). The final cost included only laboratory expenses (based on 2022 average consumable market prices and 2022 salaries) but did not include pathologist time for analyzing the slides.

## 3. Results

### 3.1. Cost Effectiveness

To analyze and quantify the cost differences between conventional histological methods and the multiplexing method, the results were split into three tables. Table 1 shows the cost difference for the stages performed by the histotechnologist to obtain histological slides (transfer punch cores from “donor” to “receptor” blocks, cleaning and trimming the paraffin blocks, microtome sectioning, floating the sections on the water bath, labeling, and staining the slides). One must bear in mind that in histology laboratories, quite often, the bulk of the costs are not related to consumables but to the required time to obtain the histological slides [32].

Using TMAs for multiplexing tissue samples significantly reduces the time required for sectioning the blocks (probably the most limiting factor in increasing efficiency in histopathology). An experienced histotechnologist can section 196 paraffin blocks in a minimum of 196 min, (assuming the blocks are of perfect quality) and seven TMA paraffin blocks in only 35 min. Accordingly, labor time is reduced by 82% if the TMA method is used. Considering all the stages from embedding to reading the slides, the total time needed is reduced by 77%. It takes a minimum of 323 min (more than 5 h without any time breaks) to prepare all the conventional slides and only 75 min for tissue microarray slides. The hematoxylin-eosin stain was performed manually for both methods. For the conventional method, we divided the 196 slides into eight separate regular staining racks (25 slots each). The staining time for one rack of conventional slides was almost 30 min, which means 240 min (4 h) for all eight racks. Staining the seven slides resulting from the TMA group (one incomplete staining rack) took only 30 min. The time to obtain the final slides (hematoxylin-eosin stained) is reflected very well in the final cost. It is 77% cheaper to use TMAs instead of conventional blocks because the cost to pay a histotechnologist is reduced from RON 356 to RON 82. 

Significant savings were also observed regarding consumables (Table 2). The cost decreased from RON 380.6 to RON 135.21, which amounts to almost 64.5%. Even when large format slides are used for TMAs blocks (5 times more expensive than the regular variety), the cost is considerably reduced. Fewer slides will also translate into less storage space for archiving (Figure 4). When factoring in all the components of histopathology service costs, the savings generated by TMA multiplexing amounts to an approx. 71% decrease (RON 217 vs. RON 736) (Table 3).

### 3.2. Influence of Tissue Processing Protocols on TMA Performance

A warning sign of insufficient processing (excessive residual water) is the difficulty in sectioning the paraffin blocks and sometimes visible artifacts such as shrinking of the tissue block within the surrounding paraffin, holes in the resulting paraffin sections, etc., [28]. We observed an inadequate infiltration of paraffin in the tissue samples processed with Protocol 2, most evidently for testes samples (Figure 5). Incomplete infiltration followed by shrinking of the tissue samples was less marked for the ear and skeletal muscle, while the other organs (liver, kidney, spleen, jejunum) seemed to be better processed. Additional artifacts manifested during the staining of the resulting slides, mainly for ear, skeletal muscle, and testes (Figure 6), where folds or even the complete detachment of some cores became apparent. This artifact occurred mostly with the samples from Protocol 2 and Protocol 1 (both using xylene as the clarification agent). 

For Protocol 3 and Protocol 4, the artifacts were minimal and were mainly occasional folds and wrinkles generated during the spreading of the sections on the flotation bath prior to mounting the section on the glass slides.

## 4. Discussion

In this study we compared the cost of obtaining stained microscope slides by a conventional method of producing paraffin blocks and by a novel method of multiplexing the tissue samples. The tissue microarray method significantly accelerates the preparation and examination of tissue samples and at a much lower cost, because it allows their simultaneous analysis on a single microscope slide. Even if the preparation of TMAs would require more attention and dexterity, the end result (validation and quality control of markers/antibodies/topographical staining, etc.) will be satisfying.

Historically, a fairly large variety of methods have been used to improve multiplexing in both clinical and research laboratories. Regarding clinical laboratories, the only TMA method used presently on a routine basis is the BxChip™, which demonstrates not only a cost-reduction advantage, but also a diagnostic accuracy benefit [12,13]. In the research laboratory, the multiplexing method is chosen mainly for cost saving and as a reliable quality control tool. Hundreds of markers can be tested in a single run on TMA slides for validation or internal and external quality control [30].

In this study, we did not account for the time spent by the pathologist reviewing the slides, because this depends on the type of organ as well as on the pathology investigated. Depending on the organ and pathology involved, reading the slides can take anywhere from a few seconds when the pathologist searches for the presence of a marker, to a few minutes when searching for discrete anomalies in the cells. However, it is already widely accepted that low-density TMA slides are read faster than the same number of specimens on individual slides. Branding the recipient matrices with identification codes and a grid numbering system further aids the pathologist for a quicker review of the resulting TMA slides. 

In clinical diagnosis, the savings made possible by using TMAs regarding the storage and archiving of paraffin blocks and slides are even more spectacular, considering that generally, they must be kept for a minimum of 20 and 10 years, respectively. If sectionable matrices with customized designs (according to various pathologies) are to be implemented in pathology laboratories, the costs for consumables and storage space could be reduced by more than 50%. 

In the research laboratory, the most laborious stage in using TMAs is to transfer the biopsy cores from the “donor” block into the “receptor” block, and this requires attention and a good traceability of the samples. An embedded sectionable matrix with a predefined design assures the superior traceability of the samples, and the TMA blocks can be stored for future validations. 

One drawback of our study was the use of large-format consumables (large metal molds and large format slides, etc.). The slides had to be loaded in specialty racks and stained manually because presently, regular automatic staining devices are not adapted for large-format slides. Our method can be improved and adapted for any pathology laboratory by using smaller tissue microarrays to produce standard microscope slides (26 × 76 mm) that are fully compatible with automatic stainers.

Another challenge of our study was the loss of some of the “donor” tissue cores during sectioning and staining. While the samples from Protocol 1 and Protocol 2 (with xylene) quite regularly resulted in the peeling of some of the cores from the slides (most likely because of substandard paraffin infiltration), butanol processing performed flawlessly for both processing protocols. As expected, the poor paraffin infiltration of the “donor” blocks also affected the quality of the “receptor” block (TMA), and since *n*-butanol is partially miscible with water, it also contributed to the complete dehydration of the tissue sample.

## 5. Conclusions

Using TMAs in the pathology laboratory (for clinical diagnosis or for research purposes) can lead to significant reductions in time and cost of consumables, but their use makes apparent some neglected shortcomings in the pre-analytical stages of histoprocessing. The quality of the resulting TMA blocks and slides is influenced by all upstream processes, and if one desires to merge different organs into the same sectionable matrix, proper tissue dehydration and clearing should never be overlooked. 

Storage space is decreased by up to 96% if the multiplexing method is applied. Compared with normal paraffin embedding tissue, the use of TMA in this study appeared to be relatively easy, time saving, and cost efficient.

## Figures and Tables

**Figure 1 vetsci-10-00280-f001:**
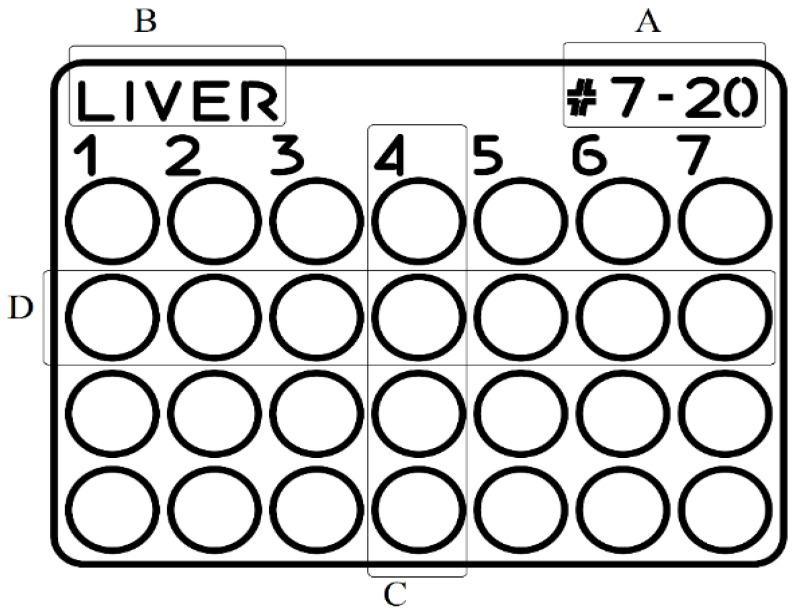
Tissue microarray design (CorelDraw Graphics Suite): (**A**) project number, (**B**) type of organ, (**C**) animal ID (columns 1 to 7), (**D**) processing protocol (four lines). The circles represent the 6 mm diameter receptacles where the tissue cores were inserted.

**Figure 2 vetsci-10-00280-f002:**
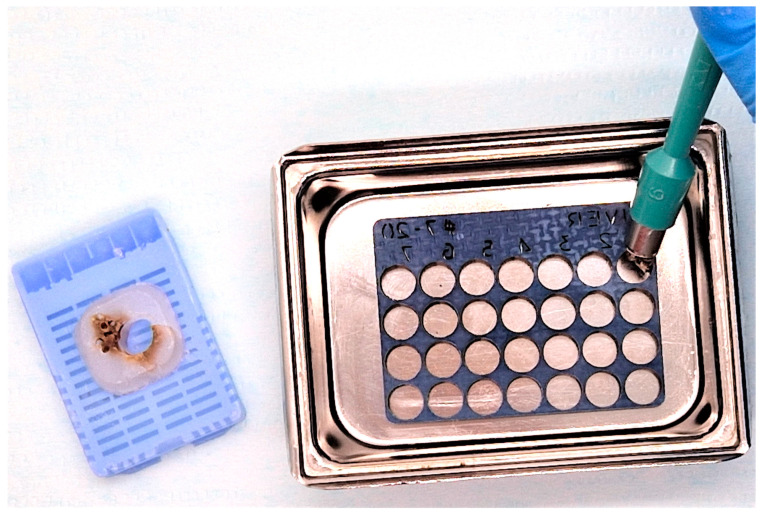
Liver tissue microarray—loading the support matrix with liver donor tissue. Tissue cores were extracted from the “donor” blocks with a 6 mm biopsy punch and arrayed in the TMA.

**Figure 3 vetsci-10-00280-f003:**
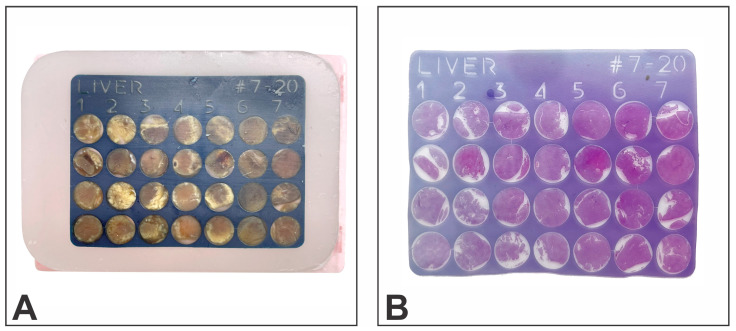
Liver tissue microarray: (**A**) sectionable matrix “receptor” block loaded with tissue cores, (**B**) hematoxylin-eosin stain.

**Figure 4 vetsci-10-00280-f004:**
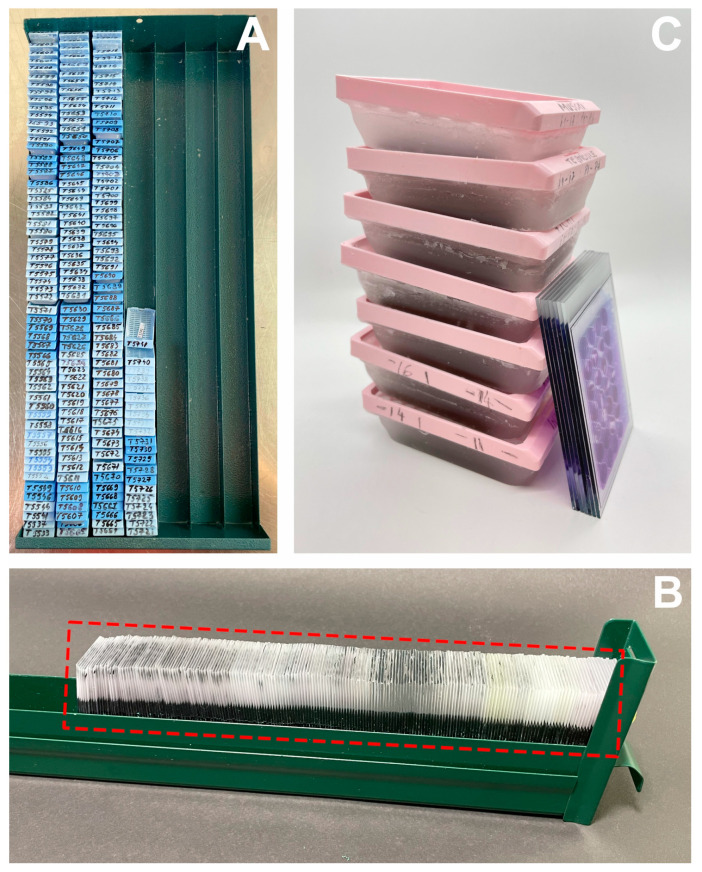
Storage/archiving: (**A**)— “Donor” paraffin blocks (196 regular cassettes), (**B**)—glass slides from paraffin “donor” blocks (196 pcs), (**C**)—seven TMA paraffin blocks (large format cassettes) and the seven resulting glass slides.

**Figure 5 vetsci-10-00280-f005:**
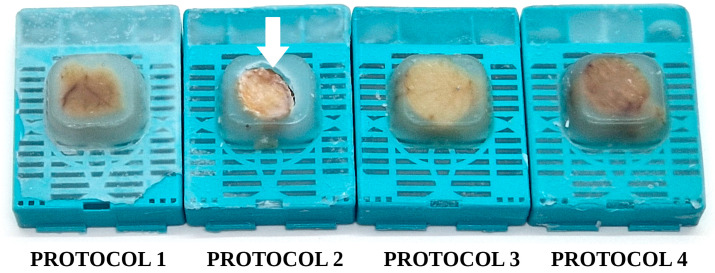
Testes “donor” paraffin blocks: Protocol 1–Protocol 4. White arrow: depression of the tissue samples within the paraffin block, a clear sign of excessive residual water/insufficient tissue processing.

**Figure 6 vetsci-10-00280-f006:**
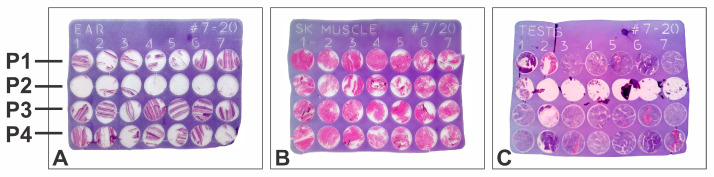
Tissue microarray, hematoxylin-eosin slide: (**A**) ear; (**B**) skeletal muscle; (**C**) testes; P1–P4: tissue processing protocols.

**Table 1 vetsci-10-00280-t001:** Time and cost required for preparing conventional versus TMA blocks and slides. All prices are based on 2022 average salaries for histotechnologists in Romania.

Stage	Salary/Minute (RON)	Conventional (196 Blocks)		TMA (7 Blocks)		Percentage Decrease	
		Time (min)	Cost (RON)	Time (min)	Cost (RON)	Time	Cost
Transfer punch cores from donor to receptor blocks	1.102	0	0.0	35	38.6	-	-
Cleaning and trimming paraffin blocks	1.102	52.27	57.6	1.87	2.1	96	96
Sectioning	1.102	196.00	216.0	35.00	38.6	82	82
Floating sections	1.102	49.00	54.0	1.75	1.9	96	96
Labeling slides	1.102	13.07	14.4	0.47	0.5	96	96
Loading racks for staining	1.102	13.07	14.4	0.47	0.5	96	96
Total Labor Costs		323.40	356.39	74.55	82.15	77	77

**Table 2 vetsci-10-00280-t002:** Comparative cost of consumables between the conventional and the TMA methods. All prices are calculated based on 2022 average consumables market prices.

Consumables	Cost per Unit/Gram (RON)	Conventional (196 Blocks)	TMA (7 Blocks)	Percentage Decrease
		Cost (RON)	Cost (RON)	Cost
TMA	15	0.00	105.00	-
Blades	3.32	63.08	9.96	84.21
Glass slide 26 × 76 mm	0.22	43.12	0.00	100.00
Glass slide 52 × 76 mm	0.803	0.00	5.62	-
Glass cover slip 24 × 50 mm	0.15	29.40	0.00	100.00
Glass cover slip 50 × 64 mm	0.84	0.00	5.88	-
Staining (reagents)	1.25	245.00	8.75	96.43
TOTAL COST		380.60	135.21	64.47

**Table 3 vetsci-10-00280-t003:** Total cost of histopathology services: conventional vs. TMA multiplexing.

Stage	Conventional (196 Blocks)	TMA (7 Blocks)	Percentage Decrease
	Cost (RON)	Cost (RON)	Cost
Labor costs	356	82	77
Consumables costs	380	135	64
Total cost	736	217	71

## Data Availability

Data will be available upon request.

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
