# Peer review of "Quality Assurance and Cost Reduction in Histopathology Laboratories Using Tissue Microarrays"

_vetsci, 2023, doi:10.3390/vetsci10040280_

Round 1

Reviewer 1 Report

The submitted manuscript entitled: »Ensuring quality and reducing costs in histopathology laboratories by using tissue microarrays« presents an alternative method for cost reduction in histopathology laboratories through the use of tissue microarrays (TMA) for parallel processing and analysis of tissue samples. The goal is to reduce time and costs and to test the effectiveness of the proposed technique with different tissue types.The presented optimization of the process is of great interest as we are always looking to reduce costs while maintaining or improving quality.

The weaknesses of the manuscript are inadequate language, lack of simple statistics, more detailed description of material and method sections, ethical issues and professional editing of the English text.

I recommend this article for publication, but there is room for substantial improvement.
after revision of the manuscript.

Abstract
According to the format and structure of the manuscript, the summary is of appropriate length, but it should be revised in terms of language. The abstract should also clearly state what methods were used.

Title

Consider this proposed title: "Quality Assurance and Cost Reduction in Histopathology Laboratories through the Use of Tissue Microarrays

Introduction
The results of other laboratories using the microarray method should be mentioned in the "Introduction" section and discussed in the "Discussion" section, since this method already existed.

Material and Methods
This section should be more clearly described, in particular it lacks information about the registration number, which should be provided since we euthanized the animals for this reason.
Add the percentage of NBF.

Results
The other missing component is the lack of statistical analysis of the data obtained, which should be added in this section as a description and in the Results section.

Discussion
As mentioned at the beginning, comparison with other publications should be added and discussed.

Author's Contribution
The author contribution does not make clear who was involved in the design of the study.

References
The references section should be revised and standardized as recommended by the journal.
A review of the English language is needed, and some minor errors should be corrected, as shown below:
- Line 45 ... a period after etc.)
- Line 70 ... long
- inserted 1.5 - 3 %
- Line 133 ... please explain what biomimetic material is
- Line 139 ... please explain what vessels are
- Line 182 ... Rephrase word "exhibited"
- In discussion (and through manuscript) decide for consistent writing minutes, hours (h)
- Start hematoyxlin-eosin with capital letters or not, there are both versions in the manuscript

Language
The manuscript should be professionally edited by an English speaker.

Reviewer 2 Report

REVIEWER COMMENTS:

1. What is the main question addressed by the research?

This study presents the results of the comparison the conventional method of producing paraffin blocks with a novel multiplexing method (tissue microarray method, TMA). The study concludes that the use of TMA appears to be relatively easy, time saving and cost-efficient.

2. Do you consider the topic original or relevant in the field? Does it
address a specific gap in the field?

In my opinion the topic of the study is relevant in the field of laboratory methods  in veterinary histopathology mad immunohistochemistry, but I estimate that it will take some time for practical reasons (education of the staff), in order the tissue microarray method (TMA)  will applied routinely  in the average vet histopathology laboratory.

3. What does it add to the subject area compared with other published material?

The study provides the basic information for the tissue microarray method (TMA) and its advantages.

4. What specific improvements should the authors consider regarding the methodology? What further controls should be considered?

I do not suggest something more.

5. Are the conclusions consistent with the evidence and arguments
presented and do they address the main question posed?

Yes, they are.

6. Are the references appropriate?

Yes, they are.

7. Please include any additional comments on the tables and figures.

Ιt would be preferable the dimensions of the images to be increased  with very detailed descriptions, in order to be understood more clearly by the readers who have no familiarity with the proposed method  till now.

Reviewer 3 Report

In this paper, authors studied tissue microarray (TMA) technology and developed a sectionable support matrix with multiple receptacles to reduce the histopathological, archiving and storage costs.

The authors collected liver, kidney, spleen, jejunum, skeletal muscle, testes and ear from New Zealand rabbits and obtained 4 pairs of biopsies, for each specimen, resulting in a total of 196 paraffin blocks.

The tissue microarray (TMA) is an important research tool in which many formalin fixed paraffin embedded (FFPE) samples can be represented in a single paraffin block, that can be used to perform immunohistochemistry to assess protein expression as well as genomic and transcriptional alterations in many samples simultaneously, thus minimizing tissue usage and reducing reagent costs.

Quantifying the cost differences between conventional histological methods and TMA method, I think the authors should include also the cost to obtain the donor block; moreover, I think that a pathologist should identify the area, where extract the tissue core and this is very expensive, in term of time consuming.

Generally, I do not agree on the comparison between normal histological technique and TMA, rather the comparison should be made, for example on an immunohistochemical investigation performed on different slides or combined into a single one, representative of numerous donor blocks.

About construction of tissue microarrays, it has not been described and characterized biomimetic material (Themis Pathology S.R.L, Bucharest, Romania). Is it 5 cm thick? How can be inserted in the  histology cassettes?

It is described that this material is etched by laser engraving to create 28 receptacles and this is another cost to describe.

The article shows the strength of describing a new method for TMA, analyzing also different protocols for tissue processing; the weakness is in the cost analysis, which in my opinion should be changed.
